# Efficacy of defocus incorporated multiple segments (DIMS) lenses and low-dose atropine on retarding myopic shift among premyopic preschoolers: Protocol for a prospective, multicenter, randomized controlled trial

**Hsin-Yu Yang**[1,2,3], **Der-Chong Tsai**[2,3]*, **Yu-Chieh Yang**[4,5], **Chiao-Yu Wang**[3], **Chia-Wei Lee**[6,7], **Pei-Wei Huang**[8]

1 Department of Ophthalmology, Taipei Veterans General Hospital Yuanshan and Suao Branch, Yilan, Taiwan, 2 National Yang Ming Chiao Tung University School of Medicine, Taipei, Taiwan, 3 Department of Ophthalmology, National Yang Ming Chiao Tung University Hospital, Yilan, Taiwan, 4 Department of Ophthalmology, Show Chwan Memorial Hospital, Changhua, Taiwan, 5 Department of Ophthalmology, Taipei Veterans General Hospital, Taipei, Taiwan, 6 Department of Ophthalmology, Fu Jen Catholic University Hospital, Fu Jen Catholic University, New Taipei City, Taiwan, 7 School of Medicine, College of Medicine, Fu Jen Catholic University, New Taipei City, Taiwan, 8 Department of Ophthalmology, Lotung Poh-Ai Hospital, Lo-Hsu Medical Foundation, Incorporation, Yilan, Taiwan

* dctsai@nycu.edu.tw

## Abstract

### Background

Myopia has been a rising problem globally. Early-onset myopia significantly increases the risk of high myopia later in life. Despite the proven benefits of increased outdoor time, optimal strategies for preventing early-onset myopia in premyopic children need further investigation.

### Methods

This randomized controlled trial aims to evaluate the efficacy of optical (Defocus Incorporated Multiple Segments [DIMS] spectacle lenses) and pharmacological (0.01% atropine eye drops) interventions in preventing myopia among premyopic preschoolers. We will recruit 234 premyopic, asymptomatic 5-to-6-year-old children who will have received cycloplegic autorefraction examination in a countywide kindergarten eye care program in Yilan County, Taiwan. Eligible participants will be randomly assigned to DIMS spectacles (n = 78), 0.01% atropine (n = 78), or usual care (n = 78). In the DIMS group, preschoolers will be instructed to wear spectacles at home before entering elementary school but to wear them all the time after school entry. In the atropine group, subjects will be given 0.01% atropine eyedrops nightly throughout the study period. All participants will be encouraged to spend time outdoors for 2 hours every day. During the 18-month study period, cycloplegic spherical

**Data Availability Statement:** No datasets were generated or analysed during the current study. All

relevant data from this study will be made available upon study completion.

**Funding:** This study was supported by grants from National Yang Ming Chiao Tung University (RD2024-003) and National Science and Technology Council, Republic of China (NSTC 113-2314-B-A49-022). This study was supported by grants from National Yang Ming Chiao Tung University (RD2024-003) and the National Science and Technology Council, Republic of China (NSTC 113-2314-B-A49-022). The funders had no role in the study design, data collection and analysis, decision to publish, or preparation of the manuscript.

**Competing interests:** The authors have declared that no competing interests exist.

equivalent (SE) refraction, axial length, and subfoveal choroidal thickness will be measured every three months, and parents-administered questionnaires regarding risk factors for myopia will be performed every nine months.

## Primary outcome

The change in mean cycloplegic SE.

## Secondary outcomes

The cumulative percentage of incident myopia, the cumulative percentage of a fast myopic shift of SE, and the changes in mean axial length.

## Other pre-specified outcomes

The time to myopia onset, alteration in subfoveal choroidal thickness, and levels of near work/outdoor activities.

## Trial registration

This study is registered at www.clinicaltrials.gov as NCT06200194.

## Conclusion

This trial will provide insights into myopia prevention strategies and inform new eye care policies for early identification and intervention in premyopic preschoolers.

## Introduction

Childhood myopia has been a significant public health concern in East Asia for decades. In Taiwan, the prevalence of myopia surged around the school entry age. Myopia has been found in 9.0% - 10.7% of preschoolers aged 5–6 years, and this rate increased to 19.6% - 19.8% among schoolchildren aged seven years [1–3]. It has been reported that juvenile-onset myopia is associated with rapid progression to high myopia in adulthood [4]. High myopia and its complications may pressure the medical care infrastructure and health insurance system significantly. Therefore, it is imperative to adopt strategies to reduce high myopia by delaying the onset of myopia and myopic shift of refractive state among young children at high risk of developing manifest myopia.

The concept of premyopia, defined as a refractive state of an eye of $> -0.50$ diopters (D) and $\leq 0.75$ D in children with other risk factors for myopia, was proposed by the International Myopia Institute in 2019 for myopia prevention [5,6]. It has been reported that mild hyperopia ($> +0.5D$ to $\leq +2.0D$), rather than emmetropia, is the typical outcome of refractive development in children [7]. Our previous study revealed that premyopia was the most common refractive status (52%) in Taiwanese preschoolers [8]. Like myopia, premyopia at preschool age was also associated with environmental factors, such as spending more time on screen-based digital devices and less time on outdoor activities [8]. Several longitudinal studies have demonstrated that the single strongest predictor for myopia onset among nonmyopic children is the baseline status of refractive error [9,10]. Young children with premyopia should be the target population for myopia prevention strategies, especially in the post-COVID-19 era. The

COVID-19 pandemic has led to changes in children's learning patterns and lifestyles, causing an increase in myopia prevalence [11–13].

The beneficial effect of increased time outdoors to retard myopia development among schoolchildren has been shown in several clinical trials [14,15]. However, In addition to spending more time outdoors, there are no guidelines or consensus regarding applying other interventions to delay the transition from premyopia to manifest myopia. In the past two decades, various interventions have emerged for controlling childhood myopia, including pharmacological approaches with low-dose atropine and optical approaches like orthokeratology, dual-focus contact lenses, and spectacles with peripheral defocus lenses [16–25]. Only a few studies specifically focused on interventions to delay myopia onset or myopic shift in premyopic children. In Hong Kong, the Low-Concentration Atropine for Myopia Prevention (LAMP2) trial validated the concentration-dependent effectiveness of low-concentration atropine in delaying myopia onset among 4- to 9-year-old children. It was found that the 0.05% atropine group had a significantly lower 2-year incidence of myopia and less myopic shift compared to the 0.01% atropine group and the placebo group [26]. In a small-sample-sized randomized study in India involving 4- to 12-year-old premyopic children, the 0.01% atropine group showed significantly less myopic progression (-0.6 ± 0.3 D vs. -1.75 ± 0.4 D) and axial elongation (0.21 ± 0.2 mm vs. 0.48 ± 0.2 mm) compared to the control group after 2 years [27]. Besides the low-dose atropine, spectacles with peripheral defocus design, such as Defocus Incorporated Multiple Segments (DIMS) lenses, represent another low-side-effect non-invasive treatment of childhood myopia. It may be a promising modality that can be promoted among the high-risk premyopic children. However, a lack of published literature on using DIMS lenses in premyopia warrants further in-depth research.

This clinical trial aims to mainly assess the effectiveness of DIMS spectacle lenses and 0.01% atropine in slowing the progression of myopic spherical equivalent SE among premyopic children before and after elementary school entrance. Because less environmental and academic pressure is presumed at preschool, premyopic children in the DIMS group of this trial will be required to wear DIMS spectacle lenses for near work at home before school entry and full-time after-school entry. This paper aims to describe the rationale, study design, and methods of this clinical trial concerning the optical (DIMS spectacle lenses) and pharmacological (0.01% atropine) approaches for premyopia around the age of elementary school entrance.

## Material and methods

### Study design

To determine the efficacy of different interventions in slowing the SE myopia shift among premyopic children around the school entrance age, we propose to conduct a prospective, randomized, multicenter clinical trial. Two hospitals will participate in this trial: National Yang Ming Chiao Tung University Hospital, and the Yuanshan Branch of Taipei Veterans General Hospital. This trial has been approved by the Institutional Review Board of National Yang Ming Chiao Tung University Hospital (NYCUHIRB No. 2023A018) and registered at www.clinicaltrials.gov as NCT06200194. Informed consent forms will be obtained from the participating children and their parents or guardians. This trial will adhere to the tenets of the Declaration of Helsinki.

### Participants' recruitment and eligibility

We intend to conduct the recruitment process from September to December 2024, during which the eye care program (the Yilan Myopia Prevention and Vision Improvement Program [YMVIP]) will take place at all kindergartens in Yilan County. The methodology of the

YMVIP, involving repeated countywide population-based cross-sectional surveys in Yilan, has been detailed elsewhere [2]. Once premyopic preschoolers are identified through the cycloplegic refraction examination of the YMVIP program, their parents or guardians will be contacted and invited to participate in this trial. The inclusion criteria for this trial are listed as follows:

- Age at enrollment: 5–6 years old (the senior grade of kindergarten)

- Cycloplegic SE of the eye with less SE (less positive or more negative refractive error): < +1.00 D and > -0.50 D

- Astigmatism: 1.0 D or less in both eyes

- Anisometropia: 1.50 D or less

- Monocular uncorrected visual acuity: 6/7.5 or better in both eyes

- Either of the parents with moderate myopia

- Acceptance of random group allocation

- Submission of complete informed consent.

    The exclusion criteria are listed as follows:

- Strabismus or any ocular motility disorder

- Any ophthalmic and systemic disorders that might affect visual functions or refractive development

- Previous treatment of atropine or other myopia control intervention

## Randomization

The eligible participants will be classified into eight strata based on three factors: baseline cycloplegic SE (+0.50 D to < +1.00 D vs >-0.50 D to +0.25 D), gender (male vs female), and parental myopia (one vs two parents). All these factors are associated with myopia development. At the baseline visits, participants in each stratum will be randomly assigned in a 1:1:1 ratio to one of three groups: treatment group 1 (DIMS spectacle lenses), treatment group 2 (0.01% atropine eyedrops), or the control group (usual care). Random allocation will be conducted using a random sequence generated in Excel, with block randomization applied using a block size of 3.

## Intervention

Intervention measures include wearing the DIMS spectacle lenses in a stepwise pattern, receiving 0.01% atropine sulfate (0.5-mL unit concentration, preservative-free) once nightly, and maintaining usual care, such as 120 minutes of outdoor activity every day. For all participants, the study staff will emphasize the critical role of increased outdoor time in myopia prevention at each visit. The subjects' lifestyle and behavior, including outdoor time and time spent on near work, will be recorded in a diary and questionnaire.

   **DIMS spectacle lenses (Treatment group 1).**   The DIMS lenses (MiyoSmart, Hoya, Tokyo, Japan) are custom-made plastic spectacle lenses [26]. In this trial, the central optical zone of the DIMS lenses is designed to have no power since the eligible participants have normal uncorrected visual acuity. Unlike the full-time wearing of DIMS spectacle lenses by myopic children, the eligible premyopic participants in this group will be instructed to wear DIMS

spectacle lenses part-time until elementary school entrance or until a myopic shift of SE equal to or more than 0.50 D, whichever comes first. In part-time mode, subjects are asked to wear DIMS spectacle lenses after school hours and during weekends and holidays, especially when doing near work. Wearing the spectacles during school hours is not compulsory in the part-time mode. After either entering elementary school or experiencing a myopic SE shift of 0.50 D or more in either eye, the participants will transition to wearing the DIMS spectacles full-time, except during sleep and showers, until the end of the trial. At spectacle dispensing and at each follow-up visit, study staff will provide face-to-face instructions to the subjects and their parents, guardians, or caregivers about the details of spectacle care and the wearing regimen. The use of topical atropine of any concentration will not be permitted for any subject in the DIMS group during the study period.

**0.01% Atropine eyedrops (Treatment group 2).** Preservative-free 0.01% atropine eye-drops (AIMedicine, Taipei, Taiwan) will be given to the eligible subjects, with one drop installed in each eye one night before sleeping. If subjects experience significant glare or if their parents are concerned about excessive light exposure during outdoor activities, photo-chromatic glasses will be offered. Additionally, progressive glasses will be provided if subjects have difficulty with near vision.

**Usual care (Control group).** In addition to regular eye examinations every three months, study staff will provide face-to-face instructions to the subjects and their parents, guardians, or caregivers at each visit about the importance of adhering to eye care instructions, such as ensuring 120 minutes of outdoor time daily.

## Outcomes and follow-up schedule

In myopia development, cycloplegic SE and axial length are considered the most relevant and reproducible indicators for monitoring refractive changes among young children. Additionally, several factors are associated with myopic shift of SE. Accordingly, we will evaluate primary, secondary, and exploratory outcomes throughout the study period based on the follow-up schedule (Figs 1 and 2).

Suppose the cycloplegic SE reaches -1.00 D or less in either eye in treatment group 1. In that case, the DIMS spectacle lenses will be replaced with an updated prescription to maintain a corrected visual acuity of 6/7.5 or better. If the same situation happens in treatment group 2 or the control group, single-vision spectacle lenses will be provided with a distance prescription to maintain a corrected visual acuity of 6/7.5 or better.

**Primary outcome.** The primary outcome is the change in mean cycloplegic SE between the baseline visit and subsequent 3-month visits over the study period in each group.

**Secondary outcomes.** The secondary outcomes include the cumulative percentage of incident myopia, the cumulative percentage of fast myopic shift of SE, and the change in mean axial length between the baseline visit and subsequent 3-month visits over the study period in each group.

**Exploratory outcome.** Lifestyle/behavior, such as near work time and outdoor time, are potential confounding factors for myopia development. Eye care instruction may impact near work time and outdoor time. We will evaluate the changes in these factors over the study period and include them as the exploratory outcomes. Besides, time to myopia onset and changes in mean subfoveal choroidal thickness over the study period will also be exploratory outcomes.

**Definition of outcome.** According to recent large-scale epidemiological studies on childhood myopia and the definition proposed by the International Myopia Institute [5,6], myopia is defined as SE $\leq -0.5$ D and high myopia as SE $\leq -6.0$ D when ocular accommodation is

| | Enrolment | Allocation | STUDY PERIOD | | | | | | Close-out |
|---|---|---|---|---|---|---|---|---|---|
| | | | Post-allocation | | | | | | |
| **TIMEPOINT** | **-t₁** | **0** | **2nd wk (± 3d)** | **3rd mth (± 14d)** | **6th mth (± 14d)** | **9th mth (± 14d)** | **12th mth (± 14d)** | **15th mth (± 14d)** | **18th mth (± 14d)** |
| **ENROLMENT:** | | | | | | | | | |
| *Eligibility screen* | X | | | | | | | | |
| *Informed consent* | X | | | | | | | | |
| *Allocation* | | X | | | | | | | |
| **INTERVENTIONS:** | | | | | | | | | |
| *DIMS spectacle lenses (Group 1)* | | ▬▬▬ | ▬▬▬ | ▬▬▬ | ▬▬▬ | ▬▬▬ | ▬▬▬ | ▬▬▬ | ▬▬▬ |
| *0.01% Atropine eyedrops (Group 2)* | | ▬▬▬ | ▬▬▬ | ▬▬▬ | ▬▬▬ | ▬▬▬ | ▬▬▬ | ▬▬▬ | ▬▬▬ |
| *Usual care (Control group)* | | ▬▬▬ | ▬▬▬ | ▬▬▬ | ▬▬▬ | ▬▬▬ | ▬▬▬ | ▬▬▬ | ▬▬▬ |
| **ASSESSMENTS:** | | | | | | | | | |
| *Tolerability assessment* | | | X | X | X | X | X | X | X |
| *Treatment adherence assessment* | | | X | X | X | X | X | X | X |
| *Questionnaire for demographic data, past medical history, parental history* | | X | | | | | | | |
| *Questionnaire for visual habits and behaviors* | | X | | | | X | | | X |
| *Uncorrected visual acuity* | | X | X | X | X | X | X | X | X |
| *Best corrected visual acuity (if needed)* | | X | X | X | X | X | X | X | X |
| *Stereopsis: NTU-random dot* | | X | X | X | X | X | X | X | X |
| *Noncycloplegic autorefraction* | | X | X | X | X | X | X | X | X |
| *Cycloplegic autorefraction* | | X | X | X | X | X | X | X | X |
| *Ocular alignment and movement test (cover and uncover test, phoria, eye movement)* | | X | X | X | X | X | X | X | X |
| *Eye exam (slit lamp, intraocular pressure, fundus exam)* | | X | X | X | X | X | X | X | X |
| *Biometry (Axial length, keratometry, anterior chamber depth)* | | X | | X | X | X | X | X | X |
| *Optical coherence tomography (subchoroidal thickness)* | | X | | X | X | X | X | X | X |

*Abbreviation: d:day; wk: week; mth: month.

**Fig 1. Schedule of assessment and examination items.**

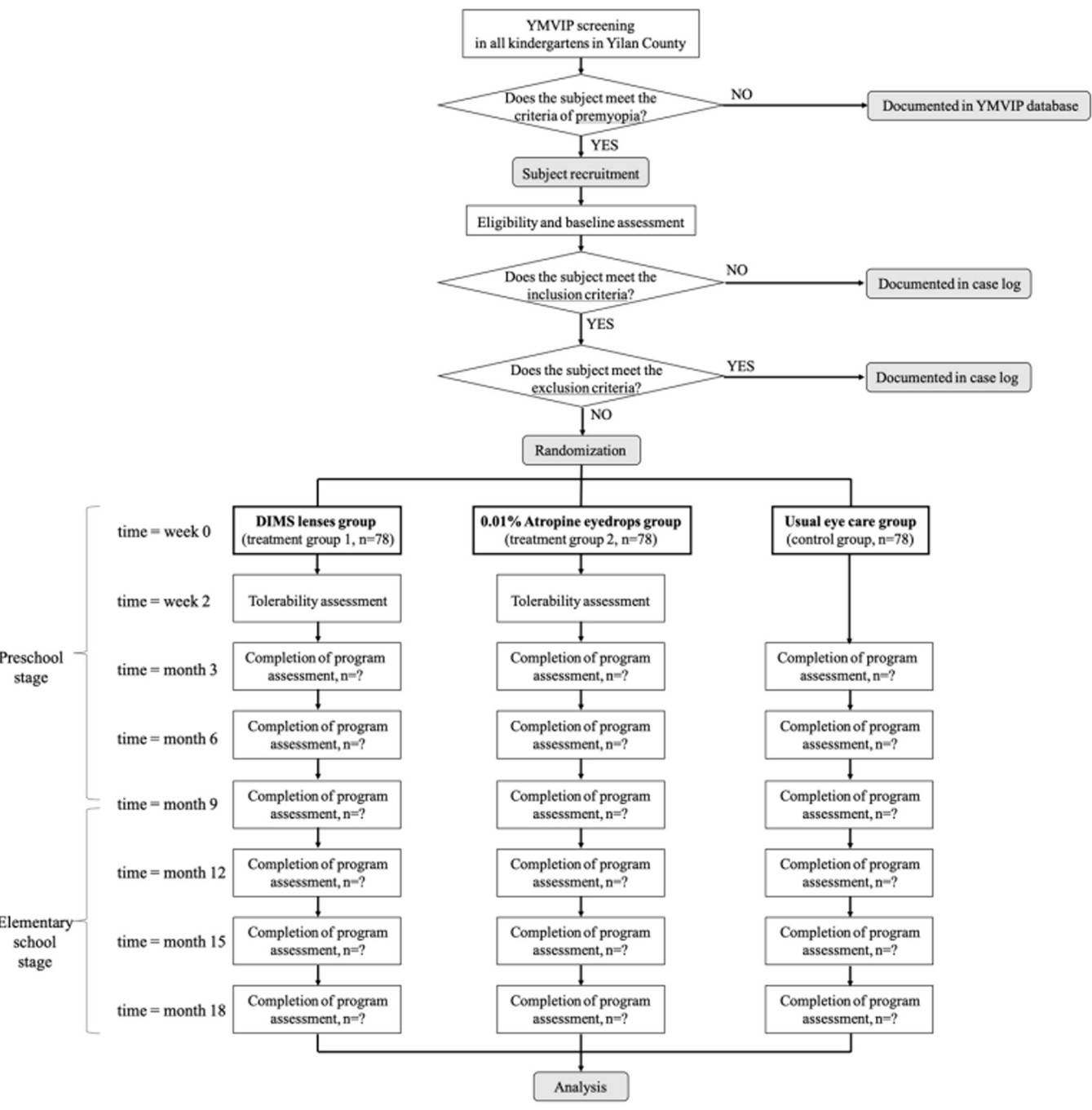

**Fig 2. Flow diagram of study design.**

relaxed. Subjects with SE $\leq-0.5$ D in either eye are classified as having myopia and those with SE $\leq-6.0$ D in either eye as having high myopia. A fast myopic shift of SE is defined as the myopic shift of SE $\geq0.75$ D over the first nine months and $\geq1.50$ D over 18 months. To avoid binocular interaction bias, only the eye with less SE at baseline (less positive or more negative refractive error) in each child was included to analyze study outcomes related to SE and axial length changes.

**Adverse event.** In the DIMS group, the main adverse event will likely be peripheral blurred vision, dizziness and eyestrain after wearing spectacles. The main adverse event in the 0.01% ATR group will likely be photophobia, blur near vision, and allergic conjunctivitis. Study staff will record any adverse event during the study period and promptly address the subjects' signs and symptoms.

**Follow-up schedule.** In addition to the baseline visit, all participants will be arranged to have follow-up visits at month 3, month 6, month 9, month 12, month 15, and month 18 after randomization (Fig 1). The subjects in the treatment groups will have additional visits for tolerance assessment in week 2. Most participants will enter elementary school after the first nine months of the follow-up period. The differences in variables of interest between the baseline and each follow-up visit will be analyzed.

## Eye examinations

At baseline visit and subsequent 3-month follow-up visits over the study period, all participants will receive the eye examination, including assessments of uncorrected/corrected visual acuity, external eye, ocular alignment, extraocular muscle motility, axial length measurement, choroidal thickness measurement, refractive status before and after cycloplegia, and a dilated fundus examination.

**Assessment of non-cycloplegic autorefraction, visual acuity, and stereopsis.** The refractive status will be first measured with a table-mound autorefractor (Topcon KR-1, Tokyo, Japan). At least three autorefraction readings will be acquired and averaged. The study staff will assess uncorrected visual acuity monocularly using a Snellen eye chart at six meters. When the right eye is tested, the left eye is covered using an eye occluder and vice versa. If the children already have the DIMS spectacles at follow-up visits, visual acuity will be assessed again when they wear their spectacles. Binocular stereopsis assessment will be carried out by study staff using NTU random-dot stereogram.

**Assessment of ocular alignment, extraocular muscle motility and intraocular pressure.** Ocular alignment will be assessed with cover and uncover test. Eye movement and ocular stability will be evaluated as well. Intraocular pressure will be measured with non-contact tonometry.

**Assessment of external eye and anterior segment.** The conjunctiva, cornea, anterior chamber, pupil size, iris, lens and anterior vitreous will be assessed using slit-lamp biomicroscopy. Children with acute keratoconjunctivitis will not be allowed to receive cycloplegic refraction, ocular biometry, or fundus examination.

**Pupil dilation and cycloplegic autorefraction.** Three drops of 1% tropicamide (administered five minutes apart) will be instilled to induce cycloplegia. Cycloplegia is presumed complete when the penlight cannot constrict the pupil anymore. If the pupil still responds to penlight stimulation, one more cycloplegic eyedrop and an additional 10-minute wait will be required before performing cycloplegic autorefraction with a table-mound autorefractor (Topcon KR-1, Tokyo, Japan). The refractive error will be assessed with at least three consecutive autorefraction readings from each participant.

**Ocular biometry measurement.** Axial length and other anatomical measurements of the eye, such as keratometry and anterior chamber depth will be carried out with an optical biometer (Lenstar LS 900, Haag-Streit, Verkauf, Switzerland).

**Choroidal thickness measurement.** Subfoveal choroidal thickness will be measured with optical coherent tomography (AgnioVue; Optovue, Fremont, CA, USA).

**Dilated fundus examination.** After pupil dilation, vitreous body, optic nerve head and posterior pole will be examined with binocular indirect ophthalmoscope.

## Questionnaire survey and compliance assessment

To collect information on potential myopia-related risk factors, the parent-administered questionnaire and eye examinations for all participants at baseline and 9- and 18-month follow-up visits will be conducted. This questionnaire comprises four sections and 16 questions regarding demographic data, parental myopia, medical information, and out-of-school behaviors such as time spent on near work, outdoor activities, sleeping and after-school care service or tutorial programs. These questions are closed-ended and have responses of yes/no options or a list of ordered choices. The respondents were asked to check the choice they felt was the most proper. Table 1 summarizes the survey questions in the questionnaire.

An unpublished study has been conducted among 40 six- to seven-year-old children (22 males and 18 females) and their parents to evaluate the concurrent validity (the agreement between children and parents) and test-retest reliability (parents' reproducibility estimates) of the questionnaire survey described above. For lifestyle, near work, outdoor activity, attendance of after-school care/tutoring activity, the five-day test-retest reliability was examined using McNemar's tests (all P > 0.05), and the measure agreement between children and their parents was examined using Cohen's kappa coefficient (0.68–0.81), which provide the evidence of adequate reliability and validity of this questionnaire.

In addition to the questionnaire, we will give each participant's parents or caregivers a diary booklet and urge them to record school attendance during the school day and amount of after-school near work activity and outdoor activity. In the treatment groups, atropine eyedrop use or DIMS spectacles wear will also be recorded in the diary daily. Treatment compliance and adherence level will be assessed based on the number of days with atropine use/DIMS spectacles worn as recorded in the dairy. Participants will be considered to have good treatment compliance if the number of days with treatment adherence is more than 80% of the total days that the participants stay in this trial.

## Sample size calculation

According to previously randomized clinical trials, the 1-year mean change of cycloplegic SE with the DIMS spectacles, 0.01% atropine eyedrops, and the control group (single vision spectacles or placebo eyedrops) was -0.17 D, -0.38 D and -0.55 D, respectively [25]. The standard deviation (SD) of mean SE changes was around 0.5 to 0.7 in the abovementioned groups, so we choose 0.6 as the SD factor. To achieve 80% power, a significance level of 0.05, 95% CI; two-sided), and 1:1:1 allocation in a test of analysis of variance (ANOVA), the estimated sample size would be 174 participants (58 in each group) based on the calculation using the software G*Power 3.1. With the estimated dropout rate of 25% over the 1.5-year follow-up, a final sample size is 234 participants (78 in each group).

## Statistical analysis

The database of examination parameters and questionnaire items of this study will be constructed with Microsoft Excel software (Redmond, WA, USA). Only the data from the eye with less SE (more myopic or less hyperopic) in each child will be used for identifying refractive status and ocular parameter analysis. Continuous variables will be expressed as mean (SD). Categorical variables will be expressed as frequency and percentage. The independent Student's t-test, ANOVA, or Pearson's chi-square test will be adopted to compare the differences between groups in terms of refractive error, demographic characteristics, and questionnaire items. Repeated measures ANOVAs will be used to test the interaction effect of time and treatment for the repeated measures during the follow-up examinations. Statistical analysis

**Table 1. Survey questions and responses in the questionnaire.**

| **Caregiver's information** |
| --- |
| • Your relationship with the child: □mother, □father, □paternal grandparent, □maternal grandparent, □others______ <br> • Age <br> • Gender <br> • Occupation |
| • Education level: <br> ➢ Child's father: □primary school, □junior high school, □senior high school/vocational school, □junior college or university, □graduate school or above <br> ➢ Child's mother: □primary school, □junior high school, □senior high school/vocational school, □junior college or university, □graduate school or above <br> • Do the parents have myopia? □no_ none has myopia, □yes_ only father has myopia, □yes_ only mother has myopia, □yes_ both have myopia |
| **Medical history of the child** |
| • Does your child have past ocular history? □no, □yes <br> ➢ If yes, what is the disease? □congenital glaucoma, □congenital cataract, □retinopathy of premature, □strabismus, □asthma, □ocular trauma, □ocular surgery, □others______ |
| • Did you have your child's eyes examined by an ophthalmologist over the past year? □no, □yes |
| • Has your child received any ophthalmic treatment over the past year? □no, □yes <br> ➢ If yes, what kind of the treatment? □cycloplegic agents, □spectacles, □orthokeratology, □patching for amblyopia |
| **Lifestyle, near-work habits and outdoor activity of the child** |
| • In recent one week, how much time a day did your child spend on doing homework (such as reading, writing, drawing and playing musical instruments)? <br> ➢ On weekdays: □none, □< 30 minutes, □≥ 30 minutes but < 1 hour, □≥ 1 but < 2 hours, □≥ 2 but< 4 hours, □≥ 4 hours <br> ➢ On weekends: □none, □< 30 minutes, □≥ 30 minutes but < 1 hour, □≥ 1 but < 2 hours, □≥ 2 but< 4 hours, □≥ 4 hours |
| • In recent one week, how much time a day did your child spend on using screen-based devices (such as watching television and playing smartphones, computers, tablets or video games)? <br> ➢ On weekdays: □none, □< 30 minutes, □≥ 30 minutes but < 1 hour, □≥ 1 but < 2 hours, □≥ 2 but< 4 hours, □≥ 4 hours <br> ➢ On weekends: □none, □< 30 minutes, □≥ 30 minutes but < 1 hour, □≥ 1 but < 2 hours, □≥ 2 but< 4 hours, □≥ 4 hours <br> • In recent one week, how much time a day did your child spend on after-school outdoor activities (such as playing balls, swimming or cycling)? <br> ➢ On weekdays: □none, □< 30 minutes, □≥ 30 minutes but < 1 hour, □≥ 1 but < 2 hours, □≥ 2 but< 4 hours, □≥ 4 hours <br> On weekends: □none, □< 30 minutes, □≥ 30 minutes but < 1 hour, □≥ 1 but < 2 hours, □≥ 2 but< 4 hours, □≥ 4 hours <br> • In recent one week, how long did your child sleep every night? □<7 hours, □≥ 7 but <8 hours, □≥ 8 but <9 hours, □≥ 9 but <10 hours, □≥10 hours |
| **Attendance of after-school activity (after entering elementary school)** |
| • Does your child attend after-school activity after entering elementary school? □yes, □no, □not sure <br> ➢ If yes, what kind of after-school activity? □on-campus care service/club activity, □off-campus care service, □off-campus tutoring program/cram school, □off-campus compound (care and tutoring) program <br> ➢ If yes, how much time a week dose your child spend on after-school activity? □< 5 hours, □≥ 5 hours but < 10 hours, □≥ 10 hours <br> ➢ If yes, how many days a week does your child attend on after-school activity? □1 day, □2 days, □3 days, □4 days, □5 days, □6 days, □7 days <br> ➢ If yes, how long has your child attended after-school activity? □< 1 semester, □≥ 1 but < 2 semesters, □≥ 2 but < 3 semesters, □≥ 3 semesters |

will be carried out using IBM SPSS Statistics 26 (IBM Corp., Armonk, N.Y., USA). All p values < 0.05 are considered statistically significant.

For the analysis of the primary outcome, the 1.5-year myopic shift of SE is calculated as the difference in SE between visits at baseline and at Month 18. The changes in SE over the 1.5-year follow-up period will be compared using ANOVA. The post hoc test will be carried out with Bonferroni correction to minimize the potential type I error due to multiple comparisons. The efficacy of myopic shift retardation of the therapeutic intervention is determined by dividing the difference in myopic shift between the treatment group and the control group with the myopic shift in the control group. The intention-to-treat analysis will be adopted to address the issue of loss to follow-up. The missing data in the repeated measures analysis will be dealt with generalized estimating equations (GEE). With one within-subject factor (time), one between-subject factor (treatment assignment) and their interactions, GEE will be employed to determine the treatment effect on the primary outcomes adjusted for some confounding factors, including age, gender, baseline SE, number of myopic parents, time spent on near works and outdoor activities.

The secondary outcomes include the proportion of incident myopia, the proportion of fast myopic shift of SE, and the changes in mean axial length over the 1.5-year follow-up period. The analyses on the secondary outcomes will also be conducted with intent-to-treat principles. Using exact unconditional methods based on the Farrington-Manning score statistic, we will calculate the difference in the cumulative incidence of myopia among groups of different treatment assignment. As a binary outcome, the proportion of fast myopic shift of SE will be analyzed with Pearson's chi-square test to compare the difference among groups. In addition, time-dependent Cox regression will be adopted to control the time-varying covariate in the analysis of the incidence of myopia and fast myopic shift of SE. Just like the analysis of myopic shift of SE, we will use ANOVA and Bonferroni correction to compare the mean change of axial length in each group. GEE will also be adapted to the treatment effect on this secondary outcome with the adjustment of the abovementioned confounding factors.

The exploratory outcomes include time to myopia onset, the changes in mean subfoveal choroidal thickness and amounts of near work/outdoor activities over the study period. In the analysis of time to myopia onset, we will assess survival analysis using a Kaplan–Meier method, with the significance based on the log-rank test. As a continuous variable, subfoveal choroidal thickness will be assessed with GEE to compare the changes among the three groups. The differences in the questionnaire items among the three groups will be estimated with ANOVA test.

The interval between the baseline visit and the last follow-up visit before school entrance is defined as "preschool stage". The interval between the last follow-up visit before school entrance and the final visit is defined as "elementary school stage". It is estimated that there will be nine months in both stages. The changes in SE and axial length in each stage are calculated as the difference in SE and axial length between the first and the last visits in the stage. We will use a paired t-test to compare the differences in mean change of SE and axial length between preschool and elementary school stages in each group.

## Discussion

Childhood myopia prevention and control have been a persistent challenge in Taiwan, especially with the increasing use of screen-based digital devices. Preschoolers with premyopia are particularly susceptible to myopia development upon entering elementary school, where academic pressure intensifies. Additionally, the post-COVID-19 pandemic era introduces new environmental and behavioral factors that may further exacerbate myopia progression.

The effectiveness of school-based eye care programs in elementary schools and kindergartens, particularly those promoting outdoor time, has been well-documented. However, there is

insufficient research on pharmacological and optical approaches to retard myopia development among premyopic preschoolers. Our proposed project aims to contribute to the existing literature by exploring how wearing DIMS spectacle lenses in a stepwise pattern may influence myopia SE shift and onset, adding evidence on the effect of low-dose atropine (0.01%) on delaying or preventing the onset of myopia among premyopic preschoolers, and comparing changes in mean SE and axial length before and after elementary school entrance.

The study protocol will help us better understand intervention strategies for myopia prevention. It may lead to a reinvention of eye care policies in early childhood, focusing on the early identification and intervention for premyopic preschoolers.

## Supporting information

**S1 File. SPIRIT-outcomes 2022 checklist.**
(PDF)

**S2 File.**
(DOC)

## Acknowledgments

The authors would like to thank all research and clinical teams involved in recruitment, coordination and data entry for this study.

## Author Contributions

**Conceptualization:** Hsin-Yu Yang, Der-Chong Tsai, Chia-Wei Lee.

**Data curation:** Hsin-Yu Yang, Der-Chong Tsai, Yu-Chieh Yang, Chiao-Yu Wang, Chia-Wei Lee, Pei-Wei Huang.

**Formal analysis:** Hsin-Yu Yang, Der-Chong Tsai.

**Funding acquisition:** Hsin-Yu Yang, Der-Chong Tsai.

**Investigation:** Hsin-Yu Yang, Der-Chong Tsai, Yu-Chieh Yang, Chiao-Yu Wang, Chia-Wei Lee, Pei-Wei Huang.

**Methodology:** Hsin-Yu Yang, Der-Chong Tsai, Yu-Chieh Yang, Chia-Wei Lee, Pei-Wei Huang.

**Project administration:** Hsin-Yu Yang, Der-Chong Tsai, Yu-Chieh Yang.

**Resources:** Hsin-Yu Yang.

**Software:** Hsin-Yu Yang, Der-Chong Tsai, Yu-Chieh Yang, Chia-Wei Lee.

**Supervision:** Hsin-Yu Yang, Der-Chong Tsai, Chiao-Yu Wang, Chia-Wei Lee, Pei-Wei Huang.

**Validation:** Hsin-Yu Yang, Der-Chong Tsai, Yu-Chieh Yang, Chia-Wei Lee, Pei-Wei Huang.

**Visualization:** Hsin-Yu Yang, Der-Chong Tsai, Yu-Chieh Yang, Chiao-Yu Wang, Chia-Wei Lee.

**Writing – original draft:** Hsin-Yu Yang, Der-Chong Tsai.

**Writing – review & editing:** Hsin-Yu Yang, Der-Chong Tsai.

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
