## [Decision Letter · Decision Letter 0]

17 Sep 2024

PONE-D-24-26994Efficacy of defocus incorporated multiple segments (DIMS) lens es and low-dose atropine on retarding myopic shift among premyopic preschoolers: Protocol for a prospective, multicenter, randomized controlled trialPLOS ONE

Dear Dr. Tsai,

Thank you for submitting your manuscript to PLOS ONE. After careful consideration, we feel that it has merit but does not fully meet PLOS ONE’s publication criteria as it currently stands. Therefore, we invite you to submit a revised version of the manuscript that addresses the points raised during the review process.

We look forward to receiving your revised manuscript.

Kind regards,

Nick Fogt

Academic Editor

PLOS ONE

 1. When submitting your revision, we need you to address these additional requirements. Please ensure that your manuscript meets PLOS ONE's style requirements, including those for file naming. The PLOS ONE style templates can be found at  https://journals.plos.org/plosone/s/file?id=wjVg/PLOSOne_formatting_sample_main_body.pdf and  https://journals.plos.org/plosone/s/file?id=ba62/PLOSOne_formatting_sample_title_authors_affiliations.pdf.   2. Thank you for stating the following financial disclosure:   [This study was supported by grants from National Yang Ming Chiao Tung University　(RD2024-003) and National Science and Technology Council, Republic of China (NSTC 113-2314-B-A49-022)].   Please state what role the funders took in the study.  If the funders had no role, please state: ""The funders had no role in study design, data collection and analysis, decision to publish, or preparation of the manuscript.""  If this statement is not correct you must amend it as needed.  Please include this amended Role of Funder statement in your cover letter; we will change the online submission form on your behalf.   3. When completing the data availability statement of the submission form, you indicated that you will make your data available on acceptance. We strongly recommend all authors decide on a data sharing plan before acceptance, as the process can be lengthy and hold up publication timelines. Please note that, though access restrictions are acceptable now, your entire data will need to be made freely accessible if your manuscript is accepted for publication. This policy applies to all data except where public deposition would breach compliance with the protocol approved by your research ethics board. If you are unable to adhere to our open data policy, please kindly revise your statement to explain your reasoning and we will seek the editor's input on an exemption. Please be assured that, once you have provided your new statement, the assessment of your exemption will not hold up the peer review process.   4. Please include captions for your Supporting Information files at the end of your manuscript, and update any in-text citations to match accordingly. Please see our Supporting Information guidelines for more information: http://journals.plos.org/plosone/s/supporting-information. 

Additional Editor Comments:

Thank you for your submission - please address the comments from reviewer #1. In particular, consider applying a repeated measures ANOVA to these data as suggested. If you choose not to do so, please thoroughly explain the logic in your statistical approach.

Reviewers' comments:

Reviewer's Responses to Questions

**Comments to the Author**

1. Does the manuscript provide a valid rationale for the proposed study, with clearly identified and justified research questions?

Reviewer #1: Yes

Reviewer #2: Yes

2. Is the protocol technically sound and planned in a manner that will lead to a meaningful outcome and allow testing the stated hypotheses?

Reviewer #1: Yes

Reviewer #2: Yes

3. Is the methodology feasible and described in sufficient detail to allow the work to be replicable?

Reviewer #1: Yes

Reviewer #2: Yes

4. Have the authors described where all data underlying the findings will be made available when the study is complete?

Reviewer #1: No

Reviewer #2: Yes

5. Is the manuscript presented in an intelligible fashion and written in standard English?

Reviewer #1: Yes

Reviewer #2: Yes

6. Review Comments to the Author

You may also provide optional suggestions and comments to authors that they might find helpful in planning their study.

Reviewer #1: In this study protocol, a three-arm randomized control trial is being proposed to evaluate the efficacy of optical and pharmacological interventions in preventing myopia among premyopic preschoolers. The primary outcome is the mean change in cycloplegic spherical equivalent (SE). Secondary outcomes include the cumulative percentage of incident myopia, the cumulative percentage of a fast myopic shift of SE, and the changes in mean axial length.

Minor revisions:

1- Line 180: If block randomization will be used, indicate the block size.

2- Line 345: State the statistical testing method which achieves 80% power.

3- Statistical analysis (lines 356-366:) Repeated measures ANOVAs, testing the interactions of time by treatment effect, would be superior to standard ANOVAs and independent t-tests due to repeated testing.

4- Identify the software that will be used for statistical analysis.

Reviewer #2: Dear authors,

You presented a well designed protocol in order to assess the effect of DIMS and low dose atropine on myopic evolution. Overall, the sections of the protocol were clearly explained, along with the endpoints and the statistical methods you will employ. I have no specific complaints regarding the protocol, which appears detailed and complete. Good luck for your trial.

7. PLOS authors have the option to publish the peer review history of their article (what does this mean?). If published, this will include your full peer review and any attached files.

Reviewer #1: No

Reviewer #2: No

---

## [Author Response · Author response to Decision Letter 0]

25 Sep 2024

Response to Reviewers’ Comments

Reviewer #1: In this study protocol, a three-arm randomized control trial is being proposed to evaluate the efficacy of optical and pharmacological interventions in preventing myopia among premyopic preschoolers. The primary outcome is the mean change in cycloplegic spherical equivalent (SE). Secondary outcomes include the cumulative percentage of incident myopia, the cumulative percentage of a fast myopic shift of SE, and the changes in mean axial length.

1. Line 180: If block randomization will be used, indicate the block size.

Response: Thank you for the suggestion. We have added the block size of 3 in the manuscript. Please refer to page 7, line 181.

2. Line 345: State the statistical testing method which achieves 80% power.

Response: We use analysis of variance (ANOVA) to achieve 80% power. This has been included in the manuscript. Please refer to page 16, lines 358–359.

3. Statistical analysis (lines 356-366:) Repeated measures ANOVAs, testing the interactions of time by treatment effect, would be superior to standard ANOVAs and independent t-tests due to repeated testing.

Response: Thank you for this important suggestion. We have changed our analysis to repeated measures ANOVA to better account for the repeated testing. Please see page 16, lines 372–374.

4. Identify the software that will be used for statistical analysis.

Response: The software used for the statistical analysis is IBM SPSS Statistics 26 (IBM Corp., Armonk, N.Y., USA). This has been added to the manuscript on page 16, lines 374–375.

Reviewer #2: Dear authors,

You presented a well designed protocol in order to assess the effect of DIMS and low dose atropine on myopic evolution. Overall, the sections of the protocol were clearly explained, along with the endpoints and the statistical methods you will employ. I have no specific complaints regarding the protocol, which appears detailed and complete. Good luck for your trial.

Response: Thank you for your positive and encouraging feedback on our study protocol. We are grateful for your acknowledgment of the study design and clarity in the protocol. We will continue to ensure rigorous implementation and follow-up throughout the trial. We look forward to sharing the results in due course.

---

## [Editor Report · Decision Letter 1]

15 Oct 2024

Efficacy of defocus incorporated multiple segments (DIMS) lens es and low-dose atropine on retarding myopic shift among premyopic preschoolers: Protocol for a prospective, multicenter, randomized controlled trial

PONE-D-24-26994R1

Dear Dr. Tsai,

We’re pleased to inform you that your manuscript has been judged scientifically suitable for publication and will be formally accepted for publication once it meets all outstanding technical requirements.

Kind regards,

Nick Fogt

Academic Editor

PLOS ONE

Additional Editor Comments (optional):

Thank you for addressing the reviewer comments. This paper is ready to move on to the publication phase.
---

## [Editor Report · Acceptance letter]

28 Oct 2024

PONE-D-24-26994R1 

PLOS ONE

Dear Dr. Tsai, 

I'm pleased to inform you that your manuscript has been deemed suitable for publication in PLOS ONE. Congratulations! Your manuscript is now being handed over to our production team.

Kind regards, 

on behalf of

Dr. Nick Fogt 

Academic Editor

PLOS ONE